# NIR ICG-Enhanced Fluorescence: A Quantitative Evaluation of Bowel Microperfusion and Its Relation to Central Perfusion in Colorectal Surgery

**DOI:** 10.3390/cancers15235528

**Published:** 2023-11-22

**Authors:** Norma Depalma, Stefano D’Ugo, Farshad Manoochehri, Annarita Libia, William Sergi, Tiziana R. L. Marchese, Stefania Forciniti, Loretta L. del Mercato, Prisco Piscitelli, Stefano Garritano, Fabio Castellana, Roberta Zupo, Marcello Giuseppe Spampinato

**Affiliations:** 1Department of General Surgery, “Vito Fazzi” Hospital, Piazza Filippo Muratore 1, 73100 Lecce, Italy; dugo.stefano@gmail.com (S.D.); fmanoochehri@yahoo.it (F.M.); libiamd@me.com (A.L.); willi.sergi@hotmail.it (W.S.); tizianamarchese@hotmail.it (T.R.L.M.); stefanogarritano@gmail.com (S.G.); marcellospampinato@gmail.com (M.G.S.); 2Institute of Nanotechnology, National Research Council (CNR—NANOTEC), c/o Campus Ecotekne, Via Monteroni, 73100 Lecce, Italy; stefania.forciniti@nanotec.cnr.it; 3Department of Experimental Medicine, University of Salento, 73100 Lecce, Italy; dreamfazzi@gmail.com; 4Department of Interdisciplinary Medicine, University of Bari “Aldo Moro”, Piazza Umberto I, 70121 Bari, Italy; castellanafabio@gmail.com (F.C.); zuporoberta@gmail.com (R.Z.)

**Keywords:** NIR-ICG-enhanced fluorescence, minimally invasive colorectal surgery, anastomotic leakage, colorectal cancer

## Abstract

**Simple Summary:**

Near-infrared indocyanine green (ICG)-enhanced fluorescence has been advocated as a reliable real-time technique to assess bowel perfusion during anastomosis formation in colorectal surgery. However, to date, no standardized protocols nor a quantitative imaging assessment are available. The aim of this study was to evaluate the timing of fluorescence as a reproducible, cost-effective parameter and its efficacy in predicting anastomotic leakage. The authors confirmed that patients with a longer perfusion timing, identified as delta timing, are at risk of developing anastomotic leakages. Furthermore, a real-time evaluation of a delta-timing/heart rate interaction provided a sensitive cut-off value to predict anastomotic leakage. The analysis of the timing of fluorescence can be easily applied during intraoperative ICG-enhanced fluorescence and may be used as a quantitative, objective parameter to guide surgical decision making.

**Abstract:**

Background: To date, no standardized protocols nor a quantitative assessment of the near-infrared fluorescence angiography with indocyanine green (NIR-ICG) are available. The aim of this study was to evaluate the timing of fluorescence as a reproducible parameter and its efficacy in predicting anastomotic leakage (AL) in colorectal surgery. Methods: A consecutive cohort of 108 patients undergoing minimally invasive elective procedures for colorectal cancer was prospectively enrolled. The difference between macro and microperfusion (ΔT) was obtained by calculating the timing of fluorescence at the level of iliac artery division and colonic wall, respectively. Results: Subjects with a ΔT ≥ 15.5± 0.5 s had a higher tendency to develop an AL (*p* < 0.01). The ΔT/heart rate interaction was found to predict AL with an odds ratio of 1.02 (*p* < 0.01); a cut-off threshold of 832 was identified (sensitivity 0.86, specificity 0.77). Perfusion parameters were also associated with a faster bowel motility resumption and a reduced length of hospital stay. Conclusions: The analysis of the timing of fluorescence provides a quantitative, easy evaluation of tissue perfusion. A ΔT/HR interaction ≥832 may be used as a real-time parameter to guide surgical decision making in colorectal surgery.

## 1. Introduction

In the last few decades, colorectal surgery has undergone several major changes in terms of technical improvements and perioperative management. Nonetheless, anastomotic leakage (AL) still represents one of the most dreaded complications with a huge impact on clinical practice and negative socioeconomic implications [1,2,3]. Due to the lack of a standardized definition for AL, its rate varies widely in the literature, being reported from 1% to 30% of all colorectal resections with a mortality rate ranging from 6% to 22% [4,5].

Several factors have been recognized in the pathogenesis of anastomotic complications [6], among which the hypo-oxygenation secondary to inadequate vascular perfusion of the anastomosis seems to play a capital role [7,8]. Recently, near-infrared indocyanine green (NIR-ICG)-enhanced fluorescence has been advocated as a reliable real-time technique to assess bowel perfusion during anastomosis formation [9,10,11]. Many studies demonstrate its potential benefit in preventing and reducing AL in colorectal surgery [12].

However, to date, no standardized protocols nor a quantitative imaging assessment are available. This results in a great variability of subjective interpretations, limiting the use and the diffusion of this technique in current surgical practice. Indeed, it has been demonstrated that the surgeon’s qualitative evaluation of fluorescence intensity depends on different optical and chemical parameters such as dye concentration, distance between the camera and the target, quality and speed of the camera, frequency and intensity of respiratory movements, body mass index (BMI) and hemodynamic status [13,14].

The aim of this study was to evaluate the timing of fluorescence as a reproducible, cost-effective parameter and its efficacy in predicting AL.

## 2. Materials and Methods

### 2.1. Data Collection

A consecutive cohort of 108 patients undergoing minimally invasive elective resections for colorectal cancer was prospectively enrolled from March 2021 to December 2022 at the Department of General Surgery of the “Vito Fazzi” Hospital, Lecce. In order to limit surgical bias related to interoperator variability, all the patients were operated by the same surgeon or under his direct supervision. Written informed consent was obtained from all the patients included in the study.

Inclusion criteria were: (a) patients ≥ 18 years; (b) diagnosis of colorectal cancer irrespective of tumor locations; (c) laparoscopic surgery; (d) ileo-colonic/colo-colonic/colorectal anastomosis formation.

The exclusion criteria included history of adverse reaction to ICG and/or iodine, emergency procedures, benign diseases, pregnant/lactating women, anal canal cancer, all procedures including a stoma formation without a synchronous anastomosis, and severe renal impairment (estimated creatinine clearance <45 mL/min).

Preoperative clinical features were recorded and included age, gender, smoking habits, American Society of Anesthesiologists (ASA) score, Body Mass Index (BMI), nutritional status based on albumin levels, comorbidities, previous abdominal surgery, steroid therapy, and neoadjuvant treatment. Other preoperative data recorded were history of cardiovascular disease, use of beta-blockers use and radiological evidence of critical or subcritical plaques in the abdominal aorta.

All the procedures were performed laparoscopically with the same optical system: a rigid 30°, 10 mm EndoEye 2D endoscope with a xenon light source providing both visible and NIR excitation light (3CCD Olympus Full HD, Visera Elite II, Olympus Corporation, Tokyo, Japan). In NIR excitation, all fluorescence images appeared green on a black background; the surgical monitor was a full HD 1080p (1920 × 1080) pixel, flat panel LCD, 16:9 widescreen.

Before the anastomosis, the site of colonic/ileal resection, clinically chosen by the surgeon, was marked with a clip. All the vascular supply to the bowel segment was transected in order to avoid any collateral perfusion by the Riolano arcade and exclude possible vascular backflow. Following vascular transection, the fluorophore (Verdye^®^, A.P.M. S.r.l, Milano, Italy) was diluted in 10 mL of water for injection, and a bolus of 0.2 mg/kg ICG was injected into a peripheral vein, which was followed by a flush of saline solution for 10 s. To ensure the homogeneity of endoscope–bowel distance among patients, the common iliac artery bifurcation was identified as a fixed, visual target during the fluorescence angiography. The macro and microperfusion were then obtained by calculating the timing of fluorescence at the level of iliac artery division (Ti) and colonic wall (Tw), respectively (Figure 1). All the timings were recorded, as well as the patient systolic and diastolic blood pressures, heart rate and any possible vasopressors infusion during ICG administration. In case of left-sided cancers requiring transverse/splenic flexure resection or left hemicolectomy or high rectal resection, a second angiography was performed after anastomosis completion, and the timing of bowel fluorescence was reported (Ta). In low and ultra-low rectal resections, in which the distal stump visualization in the abdominal cavity was not feasible, the anastomotic staple line and the rectal mucosa were analyzed transanally. Moreover, in all left-sided procedures, an air leakage test was performed, and the integrity of doughnuts was checked in order to identify any potential mechanical risk for AL.

Any change in surgical plan regarding the transection line and/or the anastomosis formation after ICG-enhanced fluorescence injection was recorded. Other intraoperative details recorded included operating time, associated procedures, type of anastomosis (manual/mechanical, side-to-side/end-to-end), stapler type, left colic artery preservation, intraoperative bleeding and complications, conversion to open surgery, diverting ileostomy and drainage placement.

During the postoperative course, a fast-track protocol was applied to all patients following the Enhanced Recovery After Surgery (ERAS) Guidelines [15]. Postoperative data included, among others, intensive care unit (ICU) stay, oral diet resumption, time to first flatus and stool passage, length of hospital stay (LOS), need for transfusion, complication rate according to Clavien–Dindo classification [16], reoperation rate, 60-day readmission rate and 30-day mortality rate. The anastomotic leakage was classified in three grades of severity according to the definition of the International Study Group of Rectal Cancer [17]. It was identified as any disruption of the anastomosis, including leakage, abscess and enteric fistula, verified by water-soluble contrast enema, pelvic computed tomography and clinical findings occurring during hospitalization and within 1 month after surgery. In case of diverting ileostomy, occult AL was also evaluated performing a flexible proctoscopy within 30 days from hospital discharge. Figure 2 summarizes the study design.

### 2.2. Outcomes

For each patient, the perfusion time factors were hemodynamic parameters during ICG injection (systolic/diastolic blood pressure and heart rate, SBP/DBP/HR), delta time (ΔT) and T ratio (TR), which was defined as the difference and ratio between central (Ti) and bowel timing (Tw), respectively.

The primary outcome of the study was the evaluation of the relationship between ΔT, hemodynamic parameters, TR and anastomotic leakage, trying to identify a possible cut-off value of both parameters in order to predict AL.

Secondary outcomes included the following:-To examine the correlation between AL and any change in the surgical plan (change in the transection line and/or revision of the anastomosis);-To identify any possible risk factor for AL and its relation to ΔT-HR;-To consider the main postoperative clinical outcomes (oral diet resumption, time to first flatus and stool passage, length of hospital stay, need for transfusion) in relation to ΔT-HR;-To investigate the relationship between ΔT and major postoperative complications (Clavien–Dindo ≥ 3).

### 2.3. Statistical Analysis

The entire sample included 108 patients undergoing laparoscopic bowel resection. In order to build a predictive model of anastomotic leakage and counteract the imbalance found in the sample, an oversampling procedure was adopted using the ROSE package in R software (version Studio 2023.09.1). A group of 540 oversampled subjects was divided according to the occurrence of a leak during the postoperative period. All the variables were reported as mean ± SD, median (iqr) for continuous values and as proportions for the categorical ones. The normal distributions of quantitative variables were tested using the Kolmogorov–Smirnov test. Based on the distribution of the quantitative data, a nonparametric approach was used to assess differences between the groups, using a Mann–Whitney U test for independent samples. A *p*-value less than or equal to 0.05 was considered significant.

A logistic regression model was built on all leakages (presence/absence) as the dependent variable and delta time as the regressor. Four nested logistic regression models were built on AL as the dependent variable and the interaction term as the regressor. Model 1 was a raw model in order to assess the relationship between AL and the interaction term. Model 2 was built on AL as the dependent variable and the interaction term as the regressor adjusted for sociodemographic confounders such as age, sex and BMI. Model 3 was built using model 2 regressors plus multimorbidity score. The multimorbidity score was obtained as a count of all assessed conditions present at the time of study entry with a particular attention to cardiovascular diseases. The fourth model was built using all model 3 regressors plus atherosclerosis plaques.

An ROC curve was built on the AL presence using model 3 prediction to evaluate the optimal AL prediction threshold of the model itself.

In order to investigate any relationship between the degree of postoperative surgical complications and the interaction term, a second ROC curve was built on Clavien–Dindo classification as the outcome and the delta time/heart rate interaction term as the predictor. In this regard, the Clavien–Dindo classification was dichotomized by placing class ≥3 as a cut-off value. All statistical analyses were performed using RStudio 2023.03.1 and the following packages: Tidyverse, Gmodels, KableExtra, ROSE, pROC, ggplot2, and HMISC.

## 3. Results

### 3.1. Descriptive Analysis

The authors prospectively included 108 patients with diagnosis of colorectal cancer, undergoing elective right colectomy (35.18%), left colectomy (23.14%), low anterior resection (25.92%), transverse colectomy (1.85%), sigmoid colectomy (7.40%) and splenic flexure colectomy (15.74%), respectively. The patients’ characteristics and operative data are shown in Table 1 and Table 2. The examined cohort presented a mean age of 69 ± 10.63 years, a mean BMI of 26.5 ± 3.92 and a sufficient nutritional status (mean albumin level of 3.92 ± 3 g/dL). There was a slight prevalence of men (53.7% versus 46.30%), and 6.50% of patients were current smokers, while 23.10% were ex-smokers; 21.29% of subjects suffered from cardiovascular diseases and 41.65% had CT findings of atherosclerotic disease. Among the patients with a diagnosis of rectal cancer (26.90%), more than half (19 patients) underwent a neoadjuvant treatment according to ESMO Guidelines [18] and had a loop ileostomy during surgery (16 patients).

The hemodynamic parameters during ICG injections were as follows: mean SBP 103.25 ± 15.74 mmHg, mean DBP 62.87 ± 9.82 mmHg, mean HR 65.2 ± 13.43 bpm. The mean Ti (timing of fluorescence at the iliac axis) was 32.18 ± 14.49 s, while the mean Tw (timing of fluorescence at the colonic wall) was 45.84 ± 21.01, which led to a mean delta time (ΔT: Tw − Ti) of 13.69 ± 13.12 and a mean T ratio (TR: Ti/Tw) of 0.72 ± 0.15.

In all left-sided procedures, a second fluorescence angiography was repeated, and a mean Ta (timing after anastomosis) was 42.92 ± 17.06 s.

None of the patients had a positive anastomotic air leak test nor intraoperative complications, but after NIR-enhanced fluorescence, four patients (3.70%) needed a change in the transection line, which was moved proximally in all the cases. Furthermore, in seven cases (6.48%), an anastomotic reinforcement/repair with direct suture was considered necessary. As depicted in Table 3 and Table 4, half of the patients that required a change in the transection line did experience an anastomotic complication as well as almost half of those who required an anastomotic reinforcement/repair. The two groups (patients with or without a change in the surgical strategy) showed a statistically significant difference in terms of AL with a *p*-value of <0.01 and 0.02, respectively.

As shown in Table 5, an oral diet was resumed in a mean of 2.14 ± 1.56 days and, restoration of bowel function was observed in a mean of 1.84 ± 1.02 days for gas passage, 3.18 ± 1.18 days for stool passage and 2 ± 1.24 days for liquid stool passage in stoma patients.

A total of five patients (4.62%) developed a major complication defined as Clavien–Dindo ≥3, among which three patients (2.77%) experimented a grade C anastomotic leak treated by a laparoscopic anastomosis repair with ileostomy in two cases and an anastomotic breakdown with a Hartmann procedure in one patient. In details, two patients had undergone a left colectomy, and one patient had undergone a right colectomy. No occult AL was recorded, while a patient primarily submitted to a Ta-TME with a ultra-low colorectal anastomosis was readmitted within 30 days due to a grade C anastomotic leakage caused by an ischemic colonic stump. No mortality was reported. The total AL rate was 3.7%. A total of 11 patients were treated with a total mesorectal excision with ultra-low colorectal or coloanal anastomosis. Among them, only one patient experienced a late leakage treated by readmission and reoperation, and none developed a 60-day anastomotic stricture/stenosis.

### 3.2. Quantitative Analysis

In order to create a more robust statistical model, an oversampling of the statistical sample was performed, leading to a cohort of 540 patients divided in two groups (with and without AL). Table 6 summarizes all patients’ characteristics after oversampling. The two groups showed some statistically significant differences in terms of gender, BMI, ASA score and ICG metrics. In details, in the AL group, there was a prevalence of men (73.40%, *p* < 0.01), subjects with BMI ≥ 28.53 ± 4.91 (*p* < 0.01) and ASA score ≥ 3 (*p* < 0.01). The hemodynamic parameters recorded during ICG injection showed that a lower systolic blood pressure/diastolic blood pressure (SBP: 91.07 ± 9.85 mmHg; DBP: 57.85 ± 5.81 mmHg) and a higher heart rate (73.14 ± 17.76 bpm) were associated with an AL with a *p*-value < 0.01. Moreover, a longer timing of central perfusion (Ti: 44.65 ± 24.16 s) and tissue perfusion (Tw: 58.03 ± 26.41 sec) were present in the AL group (*p* < 0.01). Consequently, subjects with a ΔT ≥ 15.52 ± 0.5 and a TR ≥ 0.73 ± 0.09 had a statistically higher tendency to develop AL (*p* < 0.01 and 0.03, respectively).

A logistic regression model was created considering ΔT alone and, subsequently, the ΔT/HR interaction (ΔT * HR) as regressors. An interaction effect between ΔT and HR was observed, and the test confirmed the impact of these metrics in predicting AL with an OR of 1.02 with a *p*-value < 0.01 for both parameters; see Table 7.

As reported in Table 8, the authors described the efficacy of this correlation with an ROC curve of the ΔT/HR interaction. A cut-off threshold of 832 was found to accurately correlate with a leakage onset (accuracy: 0.81; range: 0.78–0.85) with a sensitivity of 0.86 and specificity of 0.77. The perfusion status of patients with AL was then classified using the cut-off obtained by the ROC curve (Figure 3). All patients experimenting an anastomotic complication were distributed in an area of longer perfusion time (ΔT/HR interaction), which should be considered a risk zone for developing AL.

A descriptive analysis of the distribution of subjects with a change in the surgical strategy according to the ΔT/HR interaction revealed that only patients with a fluorescence parameter ≥832 developed a leakage, and in all of them, both a change in the transection line and a direct repair of the anastomosis was required; see Table 9.

A further logistic regression analysis was performed, and the following characteristics were statistically proved to be risk factors associated with AL (Table 10): age with an OR of 0.93, BMI with an OR of 1.17, male sex with an OR of 6.07 and multimorbidity score with an OR of 1.62.

According to a Pearson’s correlation plot shown in Figure 4, the ΔT/HR interaction had a slight correlation with the time to first flatus and the length of hospital stay. If considering the ΔT alone, also the resumption of an oral diet seemed to correlate with the perfusion parameters.

To evaluate any possible interaction between ΔT/HR rate and major postoperative morbidity (considered as any Clavien–Dindo ≥ 3), a second ROC curve was built, as shown in Table 11. The analysis confirmed the threshold of 832.5 as a reliable cut-off to predict postoperative complications with an AUC of 0.7 (0.65 to 0.74). Although the parameter has a discrete sensitivity (0.66), it showed a high specificity (0.89).

## 4. Discussion

Initially introduced in colorectal surgery by Kudszus [19] in 2010, NIR ICG-enhanced fluorescence has gained popularity among surgeons, and there is a flourishing debate in the literature assessing the benefits and limitations of this technique. Indeed, other tests have been previously proposed in order to evaluate blood flow or vascular architecture (Doppler ultrasound, laser Doppler flowmetry, angiography, and oxygen spectroscopy) [20,21], but none of them has reached consensus due to technical difficulties, elevated costs and lack of reproducibility.

Several authors advocate that intraoperative fluorescence angiography is beneficial in assessing perfusion, thus reducing the rate of anastomotic leakage and eventually lowering morbidity and mortality in colorectal surgery for cancer [22,23]. In the last few years, a great effort has been made to prove the efficacy of this technique in predicting potential anastomotic leakages and guiding the surgical plan, mainly when performing colorectal anastomoses [24,25,26,27].

Surgeons’ subjective evaluation of intestinal perfusion based on visual inspection has proven to be of poor predictive value [28]. On the contrary, ICG-enhanced fluorescence seems to be reproducible, safe and cost effective, allowing surgeons to be able to change the operative plan based on real-time imaging [29,30].

Nonetheless, to date, no Randomized Controlled Trial has really demonstrated the superiority of ICG use compared to crude visualization. Both De Nardi [31] and Jafari et al. [32] concluded that ICG NIR-enhanced fluorescence can effectively assess vascularization, although no statistically significant reduction in AL was proven. It should be remarked, however, that the PILLAR III trial was concluded early due to decreasing accrual rates, which could have adversely affected the statistical power of the analysis, and no data about the change in the transection point (considered as any possible AL avoided) were recorded. On the other hand, De Nardi found that the technique allowed for a change in the surgical management in 11% of patients with a change of the intraoperative strategy performed by the surgeon. Only the FLAG Trial [33] concluded that ICG fluorescence led to a reduction in grade A anastomotic leakages in case of low colorectal anastomoses. Despite these promising results, the trial has several limitations due to the absence of blinding, a great procedural heterogeneity and the fact that it is a single-center study.

Moreover, indocyanine green angiography has still several drawbacks linked to the huge variability of study protocols and the lack of an objective interpretation of the fluorescence data. Considering that fluorescence intensity is inversely correlated to the source-to-target distance and that perfusion is a dynamic process in which the fluorophore tends to distribute even to the ischemic zones over time, it is easy to understand that the need for a quantitative and stable metric is of utmost importance to ensure a reliable and reproducible evaluation. Quantitative analyses based on fluorescence intensity (Fmax) and fluorescence timing (Tmax, T1/2) have been carried out by Wada [34], who demonstrated that both parameters are decreased in the AL group of patients. However, analytic software is expensive and rare in surgical practice, and laparoscopic/robotic consoles integrated with augmented reality systems, as proposed by Diana et al. [35,36], are still on an experimental level. Recently, Diana et al. conducted a very interesting clinical study, proving that fluorescence-based enhanced reality (FLER) has a strong correlation with capillary lactates and the mitochondria efficiency chain. Thus, the authors concluded that this technique is reliable in correctly predicting anastomotic complications. In spite of its promising perspectives, this study has some limitations due to the small sample size, the slow recruitment, the high procedural heterogeneity, and the need for specific, expensive software to obtain the enhanced reality perfusion cartography [37]. Other promising results came from the work conducted by Xiaoming et al. who combined the ICG lymphography and ultrasonography for low-pressure vein localization. Despite the optimal results in vascular breast surgery, this technique is unlikely to be applicable in minimally invasive colorectal surgery. In fact, the dye injection should be made by means of an intraoperative colonoscopy and after a maximum of 1 min, a surgeon with high training in ultrasonosgraphy should detect the venous/lymphatic outflow on an overdistended colon [38].

In order to overcome these drawbacks, and moving from the intuition of two recent works conducted by Son [39] and Kim [40], the present study examined the interaction between fluorescence timing and heart rate as a potential analytic tool which could be superior to fluorescence intensity. While the latter one is poorly reliable and depends on several chemical and physical factors, the evaluation of flushing times is cost effective and easy to apply in the surgical setting. Moreover, to avoid any possible confounding factor, all the surgeries in the study followed a rigid protocol: all measures were performed during the laparoscopic time in the abdominal cavity, so it can be assumed that light conditions were homogeneous in all patients, the optical system and camera being equal; the distance between the laparoscopic camera lens and the target colon can be considered uniform, since both the left iliac axis and the intestinal transection point are visible at the same time. Furthermore, all surgeries were completed by the same operator or under his direct supervision, and the ICG concentration and administration were identical in all subjects. To allow the detection of any type of anastomotic defect, including occult anastomotic leakages, a 30-day proctoscopy was performed in all the patients with ileostomy.

As largely demonstrated in cardiosurgery [14], perfusion time factors reflect the patient hemodynamic status (systemic blood pressure and volume, cardiac output, vascular resistances, vasopressor agents), ensuring a complex understanding of both macro- and microperfusion. Nonetheless, to date, the only guidance in the interpretation of fluorescence timing is the cut-off value of 60 s [40]. After this time, the tissue perfusion should be considered inadequate, and a change in the operative plan should be envisaged. However, this approach seems to be simplistic to the authors, since the flushing time merely reflects colonic microcirculation and no data about central perfusion are provided.

The novelty of the present study relies on the hypothesis that microperfusion parameters need to be strictly correlated to the hemodynamic status during the intraoperative ICG angiography. After oversampling, when comparing the two groups of patients (with and without AL), a difference between central and tissue perfusion of more than 15.52 ± 0.5 s, identified as ΔT, was shown to be a significant predictor of anastomotic leakage with a *p*-value of <0.01. The time ratio ≥0.73 ± 0.09 was also statistically associated with an AL (*p* = 0.03). To support the hypothesis of the capital role of integrating the fluorescence data with a central perfusion status, the authors found that the ΔT/HR interaction (empirically calculated as the product between delta time and heart rate) is a reliable metric to accurately predict an AL with a sensitivity of 0.86 and a specificity of 0.77. They reported a cut-off threshold of 832, beyond which patients are at high risk of developing a leak with an OR of 1.02 (*p* < 0.01). This result could be of great importance for practical reasons, since it could be considered as a cut-off value capable of guiding the surgical decision making and the postoperative management.

Indeed, the authors decided to apply both perfusion parameters (TR and ΔT/HR interaction) to create a flow chart in order to classify individuals with poor perfusion. The probability of AL in the critical region (beyond 832) is higher, so it could be highly recommended to change the operative plan in order to improve the perfusion (change in the transection line, diverting ileostomy, revision of the anastomosis), and more attention should be paid by the surgical team during the postoperative course of such at-risk patients. This result confirms what was previously demonstrated by Son et al. [39].

Furthermore, it is interesting to examine that half of the patients who required a surgical change in the intraoperative strategy (anastomotic direct repair/reinforcement or change in the transection line) did eventually develop a leakage. But only those with an impaired tissue perfusion (ΔT/HR interaction ≥ 832) were at risk. This result could be explained by the multifactorial physiopathology of anastomotic leakages [41]. Still, patients with an impaired fluorescence angiography or a non-standard surgical procedure should be considered as potential fragile subjects.

The cut-off value of 832 also proved a good accuracy in predicting major postoperative morbidity (Clavien–Dindo ≥ 3) with a specificity of 0.89.

The negative effects of hypertension and atherosclerosis on vascular patency have been largely demonstrated in the literature [42,43]. By logistic regression models, the authors found a clear significant correlation between a prolonged ΔT/HR interaction and male sex, a BMI ≥28.5, age and a multimorbidity score which includes history of cardiovascular disease. Therefore, in patients with these characteristics, surgeons should maintain a high level of suspicion for AL, and any event of hypoperfusion should be avoided during surgery.

The analysis of the main postoperative clinical outcomes showed a slight linear correlation between perfusion parameters and a faster bowel mobility restoration, a quicker resumption of an oral diet and, consequently, a minor length of hospital stay. This explains how patients with a pattern of rapid perfusion have a faster recovery after surgery and better short-term outcomes.

There are several limitations to this study. First, the strength of the analyses is lowered by the small number of patients and the monocentric experience, which required an oversampling technique to be adopted. Large-scale multicenter prospective trials are needed to confirm the role of ∆T/HR interaction as a quantitative, easy measure with the potential of predicting an anastomotic leakage. Moreover, with a wider cohort of patients, subgroup analyses could be performed. Secondly, the absence of a control group does not allow making any comparison between patients performing NIR ICG-enhanced fluorescence with or without a quantitative perfusion parameter. The authors intend to verify these results in a further comparative study. At last, a wider follow-up analysis (at least 6 months) would add a better insight into the postoperative course and could help authors verify to efficacy of the ICG test in eventually predicting other anastomotic complications such as anastomotic strictures/stenosis [44].

## 5. Conclusions

In conclusion, the analysis of the timing of fluorescence can be easily applied during intraoperative ICG-enhanced fluorescence in colorectal surgery and can provide a quantitative and cost-effective evaluation of tissue perfusion. A ΔT/HR interaction ≥832 may be used as a real-time parameter to guide surgical decision making. Despite these promising results, further robust conclusions from the ongoing randomized controlled trials [45,46] are necessary.

## Figures and Tables

**Figure 1 cancers-15-05528-f001:**
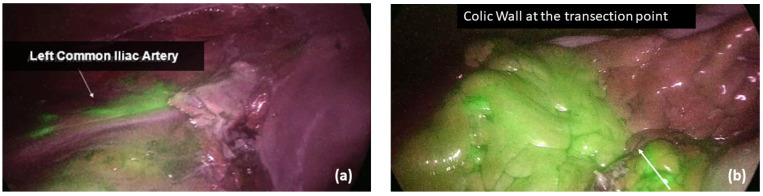
Intraoperative ICG-NIR-enhanced fluorescence; (**a**) Fluorescence at the left common iliac artery, Ti; (**b**) fluorescence at the colic wall, Tw.

**Figure 2 cancers-15-05528-f002:**
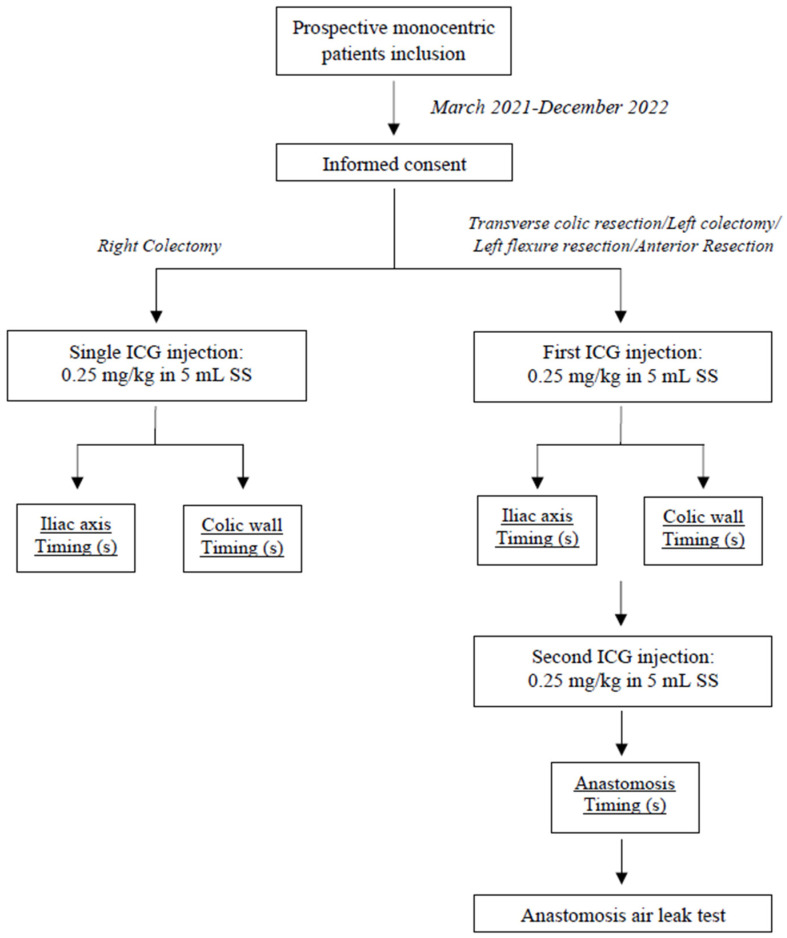
Study design.

**Figure 3 cancers-15-05528-f003:**
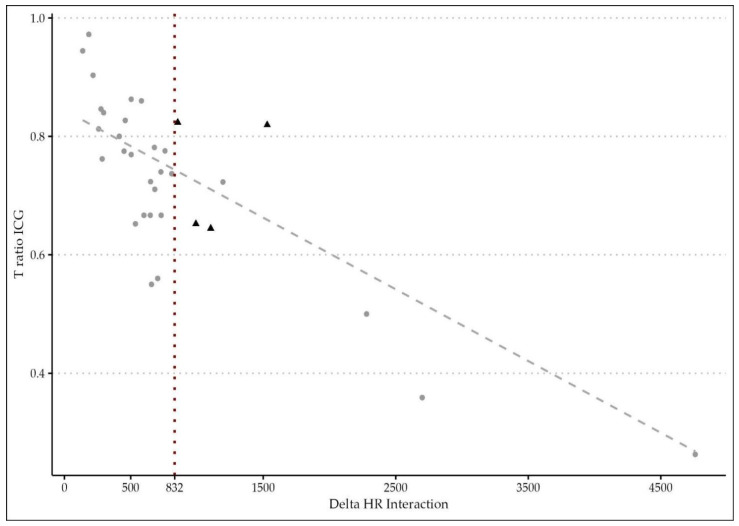
Scatter plot of time ratio and delta time/heart rate interaction on curve estimation regression analysis (N: 108). The cut-off value of 832 is indicated by a red dotted line. The sample size distribution is indicated with a gray dotted line. Patients with complications were marked with triangular shapes.

**Figure 4 cancers-15-05528-f004:**
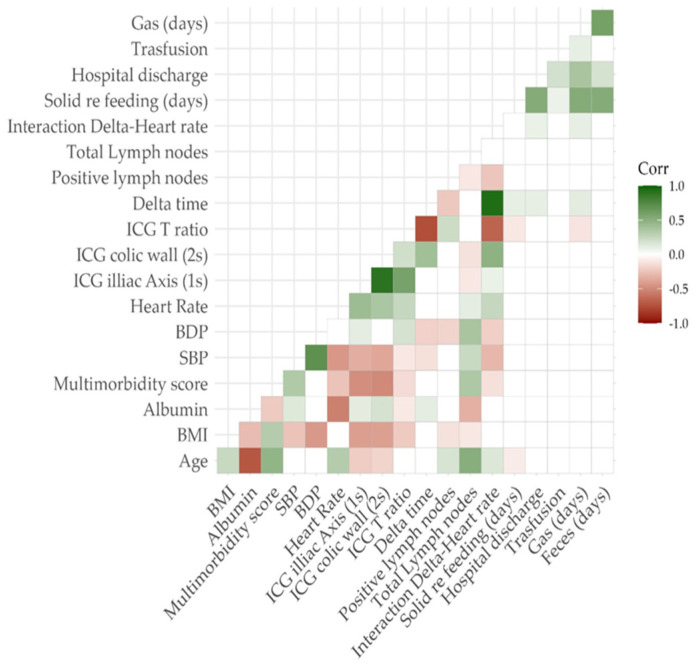
Pearson’s correlation plot.

**Table 1 cancers-15-05528-t001:** Patients’ data, N: 108.

PATIENTS’ DATA (n = 108) *		
	Mean ± SD/N (%)	Median (iqr)
**Gender**		
Male	58 (53.70)	
Female	50 (46.30)	
**Age**	69.57 ± 10.63	71 (12)
**BMI**	26.5 ± 3.92	25.7 (4.85)
**ASA**		
1	7 (6.48)	
2	51 (47.22)	
3	48 (44.44)	
4	2 (1.85)	
**Smoke Habit**		
Current smoker	7 (6.50)	
Ex-smoker	25 (23.10)	
**Albumin (g/dL)**	3.92 ± 3.00	
**Atherosclerotic disease**		
None	60 (55.55)	
Critical	8 (7.40)	
Non-critical	37 (34.25)	
**Cardiovascular diseases**		
Cardiac dysrhythmias	9 (8.33)	
Heart failure	2 (1.85)	
Coronary artery disease	6 (5.55)	
Stroke	8 (7.40)	
Valvular heart disease	7 (6.50)	
Other	2 (1.85)	
**Multimorbidity score**	1.56 ± 1.34	1 (3)
**Tumor location**		
Other than rectum	79 (73.10)	
Rectum within 15 cm from the anal verge	29 (26.90)	
**Neoadjuvant treatment**		
Radiotherapy + chemotherapy	16 (14.81)	
Chemotherapy	1 (0.92)	
Radiotherapy	2 (1.85)	

*** All data are shown as mean ± sd, median (iqr) for continuous variables and as n (%) for proportional ones.

**Table 2 cancers-15-05528-t002:** Intraoperative data, N: 108.

INTRAOPERATIVE DATA (n = 108) *		
	Mean ± SD/N (%)	Median (iqr)
**Procedure**		
Right colectomy	38 (35.18)	
Left colectomy	25 (23.14)	
Low anterior resection	28 (25.92)	
Transverse colectomy	2 (1.85)	
Sigmoid colectomy	8 (7.40)	
Splenic flexure colectomy	7 (6.50)	
**IMA low ligation**	17 (15.74)	
**Associated procedures**		
Minor hepatic resection	2 (1.85)	
Major hepatic resection	1 (0.92)	
Abdominal wall/Gerota fascia excision	4 (3.70)	
Ovariectomy	1 (0.92)	
Adhesiolysis	3 (2.77)	
Other	3 (2.77)	
**Peridural anesthesia**	37 (34.25)	
**Length of operation** (min)	212.63 ± 73.55	200 (70)
**SBP** (mmHg)	103.25 ± 15.74	100.5 (24.25)
**DBP** (mmHg)	62.87 ± 9.82	60 (14)
**HR** (bpm)	65.2 ± 13.43	63.5 (15.5)
**Ti** (s)	32.18 ± 14.49	29 (16)
**Tw** (s)	45.84 ± 21.01	40 (20.25)
**ΔT ICG** (s)	13.69 ± 13.12	10 (8)
**TR**	0.72 ± 0.15	0.74 (0.18)
**Change in the transection line**	4 (3.70)	
**Ta** (s)	42.92 ± 17.06	40.5 (17.50)
**Positive anastomotic air leak test**	-	
**Direct anastomotic repair** (suture)	7 (6.48)	
**Colorectal anastomosis: level**	54 (0.50)	
High (intraperitoneal)	27 (0.25)	
Low (extraperitoneal > 6 cm)	15 (13.8)	
Ultralow (<5 cm)	9 (8.33)	
Coloanal	3 (2.77)	
**Colorectal anastomosis: type**		
Intracorporeal	91 (84.25)	
Extracorporeal	17 (15.74)	
Side-to-side isoperistaltic	46 (42.59)	
End-to-end	42 (38.88)	
Side-to-end	20 (18.51)	
**Type of anastomotic device**		
Linear	47 (43.51)	
Circular	59 (54.62)	
Manual	2 (1.85)	
**Loop ileostomy**	16 (14.81)	
**Intraoperative complications**	-	
**Drain**		
none	77 (71.29)	
1	30 (27.77)	
2	1 (0.92)	

*** All data are shown as mean ± sd, median (iqr) for continuous variables and as n (%) for proportional ones.

**Table 3 cancers-15-05528-t003:** Distribution of AL in patients undergoing an anastomotic direct repair/reinforcement, N:108.

Anastomotic Direct Repair/Reinforcement *				
	All Leakages			
All Leakages		No	Yes	*p*-value
	No	99 (95.20)	2 (50)	0.02
	Yes	5 (4.80)	2 (50)	

* All data are shown as n (%).

**Table 4 cancers-15-05528-t004:** Distribution of AL in patients undergoing a change in the transection line, N:108.

Change in the Transection Line *				
	All Leakages			
All Leakages		No	Yes	*p*-value
	No	102 (98.10)	2 (50)	<0.01
	Yes	2 (1.90)	2 (50)	

*** All data are shown as n (%).

**Table 5 cancers-15-05528-t005:** Postoperative data, N: 108.

POSTOPERATIVE DATA (n = 108) *		
	Mean ± SD/N (%)	Median (iqr)
**One-night ICU** (number of pts)	3 (2.77)	
**Opioids**	12 (11.11)	
**Oral diet resumption** (POD)	2.14 ± 1.56	2 (1)
**Gas** (POD)	1.84 ± 1.02	2 (1)
**Stools** (POD)	3.18 ± 1.18	3 (2)
**Liquid stools from stoma** (POD)	2 ± 1.24	2 (2)
**Drain removal** (POD)	5.6 ± 3.35	5 (2)
**Transfusions**	3.27 ± 2.69	2 (1)
**Clavien–Dindo**		
1	18 (16.66)	
2	11 (10.18)	
3a	2 (1.85)	
3b	2 (1.85)	
4	1 (0.92)	
**Medical complications**		
Cardiac dysfunction and failure	3 (2.77)	
Acute renal failure	2 (1.85)	
Pneumonia and pulmonary failure	7 (6.48)	
Other	1 (0.92)	
**Surgical complications**		
Paralytic ileus	10 (9.25)	
Anastomotic leakage	3 (2.77)	
Surgical site infections	4 (3.70)	
Anemia/anastomotic bleeding	13 (12.03)	
Abdominal collection	4 (3.70)	
Anastomotic stenosis	-	
Other	3 (2.77)	
**AL: classification**		
A	-	
B	-	
C	3 (2.77)	
**Reoperation**		
Anastomosis repair with ileostomy	2 (1.85)	
Anastomosis breakdown with Hartmann	1 (0.92)	
**Postoperative stay** (days)	6.58 ± 4.65	5 (3)
**Post-discharge AL within 30 days**		
A	-	
B	-	
C	1 (0.92)	
**Occult leakage within 30 days**	-	
**Total AL**	4 (3.70)	
**30-day mortality**	-	
**60-day readmission**	3 (2.77)	

*** All data are shown as mean ± sd, median (iqr) for continuous variables and as n (%) for proportional ones.

**Table 6 cancers-15-05528-t006:** Description of the whole sample according to leakage onset (with/without), N: 540 *^,^**. Bolds means statistically significant.

	Without Leakage		With Leakage		
	Mean ± SD	Median (iqr)	Mean ± SD	Median (iqr)	*p* Value
Proportions (%)	266 (49.30)		274 (50.70)		
Age (years)	71.44 ± 9.88	73 (14)	69.18 ± 10	77 (15)	0.10
Gender					
Male	115 (43.20)		201 (73.40)		**<0.01**
Female	151 (56.80)		73 (26.60)		
Smoke habits					
Current smoker	--		--		
Ex-smoker	62 (23.30)		--		**<0.01**
Never	204 (76.70)		274 (100.00)		
BMI (kg/m^2^)	26.2 ± 4.13	25.26 (4.02)	28.53 ± 4.91	30.2 (3.09)	**<0.01**
ASA score	2.36 ± 0.59	2 (1)	3 ± 0	3 (0)	**<0.01**
Albumin (g/dL)	3.7 ± 0.51	3.68 (0.46)	3.78 ± 1.13	3.22 (1.79)	0.73
Multimorbidity score	1.65 ± 1.35	2 (3)	1.8 ± 1.33	1 (3)	0.30
Tumor location					
Rectum	68 (25.60)		68 (24.80)		0.84
Other location	198 (74.40)		206 (75.20)		
SBP (mmHg)	102.5 ± 14.65	100 (25)	91.07 ± 9.85	82 (21)	**<0.01**
DBP (mmHg)	63.24 ± 10.41	62 (12)	57.85 ± 5.81	57 (4)	**<0.01**
ICG heart rate (bpm)	66.24 ± 13.26	67 (17)	73.14 ± 17.76	69 (40)	**<0.01**
Ti (s)	29.45 ± 13.77	26 (14)	44.65 ± 24.16	29 (53)	**<0.01**
Tw (s)	42.46 ± 19.38	36 (20)	58.03 ± 26.41	45 (60)	**<0.01**
ΔT (s)	13.01 ± 15.1	9 (5)	15.52 ± 0.5	16 (1)	**<0.01**
T ratio ICG	0.72 ± 0.17	0.76 (0.16)	0.73 ± 0.09	0.65 (0.17)	**0.03**
Gas (days)	1.83 ± 0.99	2 (1)	1.85 ± 1.03	2 (1)	0.87
Stool (days)	3,16 ± 1.16	3 (2)	3.19 ± 1.18	3 (2)	0.89
Transfusion (n)	0.32 ± 1.30	0 (0)	0.33 ± 1.27	0 (0)	0.76
Postoperative stay (days)	6.59 ± 4.74	5 (3)	6.59 ± 4.74	5 (3)	0.81

*** Wilcoxon rank-sum test for continuous variables and chi-squared test for categorical ones. ** All data are shown as mean ± sd, median (iqr) for continuous variables and as n (%) for proportional ones.

**Table 7 cancers-15-05528-t007:** Logistic regression model on leakage onset as dependent variable and delta time versus delta time/heart rate interaction as regressors.

	Model 1				Model 2			
	OR	CI 95%	Stand. Err.	*p* Value	OR	CI 95%	Stand. Err.	*p* Value
Delta Time (s)	1.02	1.01 to 1.04	0.01	<0.01	0.80	0.70 to 0.90	0.06	0.02
Heart Rate (bpm)					0.98	0.95 to 1.01	0.01	0.08
Interaction					1.02	1.01 to 1.03	0.01	<0.01

**Table 8 cancers-15-05528-t008:** ROC curve of delta time/heart rate interaction on leakage onset.

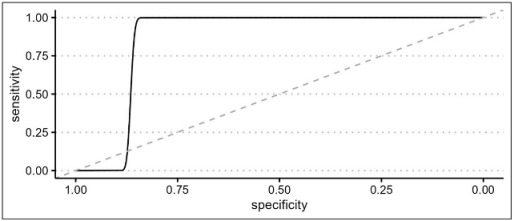				
Confusion matrix				
	Reference		AUC	0.86
Prediction	No Leakage	Leakage	Threshold	832
No Leakage	229	61	Accuracy	0.81 (0.78 to 0.85)
Leakage	37	213	Sensitivity	0.86
			Specificity	0.77

**Table 9 cancers-15-05528-t009:** Evaluation of AL and delta time/heart rate interaction in patients with a change in the surgical strategy, N: 108. We used red characters to highlight the patients who had the AL after the change in the transection line.

Delta Time/Heart Rate Interaction	Change in the Transection Line	Anastomotic Direct Repair/Reinforcement	Anastomotic Leakage
570		X	
198		X	
312		X	
**1104**	**X**	**X**	**X**
**1170**		X	
**2280**		X	
**1530**	**X**	**X**	**X**
156	X		
285	X		

**Table 10 cancers-15-05528-t010:** Logistic regression models on leakage onset as dependent variable and delta time/heart rate interaction as regressor.

	Model 1				Model 2				Model 3				Model 4			
	OR	CI 95%	Stand. Err.	*p* Value	OR	CI 95%	Stand. Err.	*p* Value	OR	CI 95%	Stand. Err.	*p* Value	OR	CI 95%	Stand. Err.	*p* Value
Delta time/heart rate	1.02	1.01 to 1.03	0.01	<0.01	1.02	1.01 to 1.03	0.01	<0.01	1.02	1.01 to 1.03	0.01	<0.01	1.02	1.01 to 1.03	0.01	<0.01
Age (years)					0.96	0.94 to 0.98	0.01	<0.01	0.93	0.90 to 0.95	0.01	<0.01	0.93	0.90 to 0.95	0.01	<0.01
Sex (female)					0.27	0.17 to 0.43	0.22	<0.01	0.15	0.08 to 0.26	0.27	<0.01	0.16	0.09 to 0.28	0.27	<0.01
BMI (kg/m^2^)					1.20	1.14 to 1.27	0.02	<0.01	1.17	1.11 to 1.23	0.02	<0.01	1.17	1.11 to 1.23	0.02	<0.01
Multimorbidity score									1.65	1.35 to 2.02	0.10	<0.01	1.62	1.33 to 1.99	0.10	<0.01
Postoperative stay (days)									0.99	0.94 to 1.03	0.02	0.65	0.98	0.94 to 1.03	0.02	0.65
Ather. plaques (critical)													0.78	0.35 to 1.74	0.40	0.55
Ather. plaques (no critical)													0.84	0.55 to 1.29	0.21	0.43

**Table 11 cancers-15-05528-t011:** ROC curve of delta time/heart rate interaction on major postoperative morbidity (Clavien–Dindo ≥ 3).

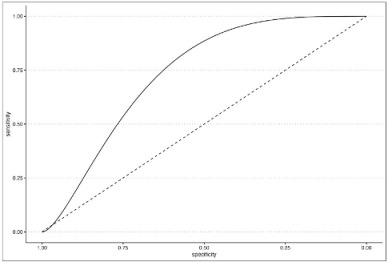	
AUC	0.70 (0.65 to 0.74)
Threshold	832.5
Sensitivity	0.66
Specificity	0.89

## Data Availability

The data presented in this study are available on request from the corresponding author. The data are not publicly available due to privacy restrictions.

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
