# Peer review of "NIR ICG-Enhanced Fluorescence: A Quantitative Evaluation of Bowel Microperfusion and Its Relation to Central Perfusion in Colorectal Surgery"

_cancers, 2023, doi:10.3390/cancers15235528_

Round 1
Reviewer 1 Report
Comments and Suggestions for Authors
This is a prospective case series evaluating the timing of ICG fluorescence in a cohort of 108 patients undergoing surgery for colorectal cancer. The aim of the study was to determine the timing of fluorescence as a reproducible parameter and its value in predicting anastomotic leakage (AL). The authors concluded that the timing of fluorescence provides a reliable quantitative parameter and a DT/HR ³ 832 may be used to guide the decision-making process during colorectal surgery.
The manuscript is well written and results are supported by adequate evidence. There are several limitations that have been correctly reported in the discussion. There is a relatively small number of patients that prevented subgroup analyses, it would be interesting to evaluate differences between patients with right, left or rectal tumors. This study Is certainly novel and can provide the basis for further research. The correlation between DT and heart rate should be further investigated to determine its accuracy in predicting AL.
I have the following minor remarks:
− In the Abstract the abbreviation NIR-ICG should be explained.
− Inclusion criteria should be clearly specified in the Methods.
Comments on the Quality of English LanguageThere are several spelling and syntax errors throughout the text.
English grammar requires revision to make the manuscript more readable.
Reviewer 2 Report
Comments and Suggestions for Authors
Norma Depalma et al. have made an interesting and clinically important study regarding a quantitative evaluation of bowel micro-perfusion and its relation to central perfusion in colorectal surgery. Their conclusion is A ΔT/HR interaction ≥ 832 may be used as a real-time parameter to guide surgical decision-making in colorectal surgery. The manuscript is well written and my only question is: “Can the authors be more specific regarding how the objective parameter guided surgical decision-making?” For example, how to intraoperatively manage the subjects subjects with a ΔT ≥ 15.5± 0.5 seconds who had a higher tendency to develop an AL (p Ë‚ 0.01)? Moreover, I would like to ask the authors to make some subset analysis regarding the grade of the anastomotic complications according to the findings of ICG test (please refer to the article from Asian fellow researchers: Asian Journal of Surgery. Volume 46, Issue 9, September 2023, Pages 3722-3726 ). Furthermore, it would be very appealing to make some discussion regarding the ICG test for low pressure vein localization, as shown in the previous publication (Liao Xiaoming et al. Asian Journal of Surgery, available online 28 August 2023: Application of combined preoperative indocyanine green lymphography and ultrasonography for low-pressure vein localization in secondary lymphedema surgery for breast cancer).
